# A Comparative Policy Analysis of Health Inequities in Access to Healthcare Across Low- and High-Income Contexts: The Cases of Pakistan and Canada

**DOI:** 10.3390/ijerph22111735

**Published:** 2025-11-17

**Authors:** Fatima Durrani, Faryal Shaikh, Mohammed Alkhaldi

**Affiliations:** 1School of Health Sciences and Psychology, Department of Public Health, Canadian University Dubai, Dubai 117781, United Arab Emirates; 20220002345@students.cud.ac.ae (F.D.);; 2The Global Health Network-The Middle East and North Africa Network (TGHN-MENA), Dubai 117781, United Arab Emirates; doctor.maidah@gmail.com; 3Army Medical College, National University of Medical Sciences (NUMS), Rawalpindi 46000, Pakistan; 4Global Health Department, Health Services Academy, Islamabad 44000, Pakistan; 5College of Medicine, Gulf Medical University, Ajman 4184, United Arab Emirates; 6School of Physical and Occupational Therapy, Faculty of Medicine and Health Sciences, McGill University, Montreal, QC H3G 1Y5, Canada; 7Centre for Tropical Medicine and Global Health, Nuffield Department of Medicine, University of Oxford, Oxford OX2 7SG, UK

**Keywords:** health inequities, social determinants of health (SDH), universal health coverage (UHC)

## Abstract

Globally, poverty remains a major obstacle to health parity, compromising well-being. This policy analysis aims to examine how poverty affects health inequities and healthcare access in two contexts: Canada, a high-income nation, and Pakistan, a low-income nation. This study employs a grounded approach, integrating a thorough review of the existing critical literature using systematic thematic analysis and synthesis. In Pakistan, chronic underinvestment, rural–urban gaps, inadequate infrastructure, and political instability exacerbate inequities in access to healthcare. Limited coverage, ineffective administrative processes, and gaps in rural healthcare delivery impede growth despite encouraging programs like the Sehat Card and the Ehsaas Program. Conversely, universal healthcare in Canada has lowered financial obstacles to access, but low-income and Indigenous communities are still impacted by service gaps, particularly in dental care, pharmacare, and mental health. Although child poverty rates have been significantly reduced by programs like the Canada Child Benefit, Indigenous children continue to endure disproportionate health risks. Findings underscore a need for equity-driven changes: Pakistan must expand rural health infrastructure and legislate health equity, while Canada should extend coverage to essential but excluded services. Findings underscore the intersecting nature of inequities driven by poverty, gender, geography, and systemic exclusion that highlight opportunities for cross-context policy learning. Canada’s equity monitoring frameworks could strengthen Pakistan’s health data systems, while Pakistan’s community-based Lady Health Worker program offers scalable grassroots models relevant for marginalized Canadian regions. Both countries must prioritize poverty alleviation as a health intervention, integrating justice, sustainability, and accountability to advance global health equity.

## 1. Introduction

While poverty is one of the major global causes of ill health, it is also a direct reflection of the improper implementation of public health policies. The Organization for Economic Co-operation and Development (OECD) has defined poverty as a lack of income and resources to support one’s living [1]. Poverty and social disadvantage remain key global drivers of health inequities. Since 1990, over 1 billion people have escaped extreme poverty, yet major inequities persist: people in the lowest life expectancy countries live 33 years less than those in the highest [2]. Impoverished populations face higher disease burdens and mortality, especially in Sub-Saharan Africa and South Asia, which account for most maternal deaths. Latin America, despite being middle-income, still has extreme poverty and inequality [2].

Poverty limits access to healthcare and basic needs such as clean water, sanitation, and nutrition, which severely impacts well-being. This is especially reflected in low-income countries such as Pakistan, where the poverty rate of 39.8% [3] significantly affects the quality of life that these groups experience. For example, the highest rates of malnutrition, high morbidity, and high mortality rates due to infectious diseases are all encountered by these groups. In addition, Ahmed et al. reported that a review of gender and health policy in Pakistan found that women had little decision-making power about using health care services due to certain societal norms, including male dominance [4]. These factors highlight the gender inequities along with socioeconomic inequities, which impact health outcomes.

In contrast, even high-income countries, such as Canada, face health inequities. The poverty rate in Canada is 9.9% [5]. Although most citizens experience fewer poverty-derived health complications, certain populations, including low-income households, and Indigenous peoples, still experience barriers to care and worsening health outcomes (e.g., heart disease, poor oral health, and shorter life expectancy). More specifically, women in these groups are more likely to live in poverty and face gaps in access and quality of care [6]. This highlights the intersection of gender and socioeconomic inequities, as well as their impact on healthcare status.

The WHO links education, housing, and sanitation to decades-long life expectancy gaps [2], while the World Bank highlights poverty in Sub-Saharan Africa and conflict zones lacking basic services [7]. The UNDP’s multidimensional poverty index shows how overlapping deprivations in health, education, and living standards harm well-being [8]. All three agencies emphasize the vicious cycle between poverty and social determinants and call for multisectoral, SDG-aligned action to promote health equity [2].

The primary aim of this study is to assess how poverty and other determinants of health drive health inequities in both high-income and low-income contexts, using Canada and Pakistan as comparative case studies. These two countries were chosen as they represent contrasting socioeconomic and health system contexts across the global income spectrum. Pakistan, a lower-middle-income country, faces challenges of limited infrastructure, underinvestment, and administrative fragmentation, while Canada, a high-income country with a universal healthcare system, continues to experience inequities rooted in income and geography. Comparing these two nations allows exploration of how distinct socioeconomic systems generate and sustain inequities, revealing both shared structural barriers and context-specific policy lessons. Although prior research has examined national health inequities, few studies have compared countries with contrasting governance and health system models to understand how structural and policy differences shape healthcare access. This study aims to systematically analyze how poverty-driven inequities influence healthcare access in Pakistan and Canada and to identify transferable policy lessons for promoting equity in both low- and high-income settings. By drawing from peer-reviewed evidence and cross-national policy analysis, the study contributes to a growing body of literature emphasizing that reducing poverty is inseparable from advancing health equity. Moreover, the findings underscore that inequities are not singular but intersecting, shaped by overlapping dimensions of gender, geography, and income. Recognizing these intersections is essential for designing inclusive, equity-driven health policies that address the compounded disadvantages experienced by women, rural populations, and Indigenous communities. This analysis applies theoretical frameworks to critically examine the effectiveness of poverty alleviation and health policies in each country. By identifying policy strengths and weaknesses, the study seeks to provide evidence-based insights for context-sensitive health equity reforms to promote better access to healthcare.

## 2. Materials and Methods

This study employs a grounded approach, integrating a thorough examination of existing literature. This approach was used to develop an understanding of how poverty-driven inequities affect access to healthcare in Canada and Pakistan. Instead of relying on a predefined framework, data from peer-reviewed publications and policy documents and reports were analyzed inductively through open, axial, and selective coding. This process allowed key themes and relationships to emerge directly from the data. Using the constant comparative method, patterns from both countries were examined to identify shared and contrasting mechanisms influencing health inequities. The approach enabled the study to build a context-sensitive, comparative framework explaining how social and policy factors shape healthcare access in different national settings of Canada and Pakistan.

The study employed a two-step purposive sampling approach. In the first step, Pakistan and Canada were selected as case study countries to provide a comparative perspective on poverty, inequity, and access to healthcare. In the second step, 29 publications, articles, and reports were purposely selected for inclusion in the review based on their direct relevance to the research topic and their publication date between 1994 and 2025. The review process consisted of a thematic synthesis of these 28 sources. The search strategy included multiple databases and platforms: Google Scholar; official government websites such as the Government of Canada, Pakistan Bureau of Statistics, Statistics Canada, and the National Institute of Population Studies (Pakistan); international organizations including the World Bank, WHO, UNDP, and OECD; global data platforms such as Data Commons and MacroTrends; and research and policy institutions including the Canadian Centre for Policy Alternatives, Stanford University’s Center on Democracy, Development and the Rule of Law (CDDRL), and Authorea. The methodology was organized into three phases: (1) initial exploration and selection of literature based on precise inclusion criteria, prioritizing sources directly relevant to the research topic; (2) rigorous data extraction focusing on key elements of poverty, healthcare access, and policy contexts; and (3) comprehensive synthesis and thematic analysis of the extracted data to generate coherent findings for comparative analysis.

This approach ensured a systematic and transparent process, providing a solid foundation for comparative insights between the two countries. The thematic synthesis process involved a purposive assessment of these selected publications, employing critical appraisal techniques. To reduce bias and promote credibility, reliability, and generalizability, different strategies were used by diversifying sampling and selection procedures of publications from diverse sources, applying clear inclusion/exclusion criteria with justification to avoid over-reliance on a single perspective, and involving multiple evaluators (authors), where two evaluators (F.D. and M.) independently screened, selected, and reviewed the literature under the guidance of third evaluator (M.A.) to resolve disagreements through discussion or consensus. This method ensured the presentation of pertinent findings supported by the authors’ insights, in line with the overarching objective of the review on poverty and health inequity in both contexts. The synthesis process was executed using MS Word and Excel programs, guaranteeing accuracy and consistency across all stages.

## 3. Results

### 3.1. Historical Evolution of Poverty Alleviation and Health Policies in Pakistan and Canada

Pakistan and Canada’s health and poverty reduction programs have evolved along different historical trajectories shaped by their socioeconomic contexts. After Pakistan’s 1947 independence, early efforts focused on land reform and rural development. In Canada, the Medical Care Act of 1966 built on earlier initiatives, culminating in the 1984 Canada Health Act, which reinforced universal, accessible, and comprehensive public healthcare. Budget cuts in the 1980s–1990s reduced social investment, but programs like the Canada Child Benefit (2016) and expanded access to dental and pharmaceutical care reflect ongoing efforts to tackle poverty and health inequities. Despite progress, Canada still faces challenges in ensuring equitable healthcare, especially for Indigenous populations [9]. These trends show Pakistan is still pursuing integrated poverty-health strategies, while Canada has established a broad welfare system.

### 3.2. Health System and Inequality Landscape

The poverty landscape of Pakistan and Canada reflects stark differences in their social and economic contexts, influencing how poverty affects health outcomes. According to Conde et al. [10], in Canada, approximately 10.1% of citizens live in poverty, an indicator of issues stemming from affordability and social exclusion of minorities such as lower-income groups. While in Pakistan 22% of the population is multidimensionally poor and is facing deprivation in health, education and living standards [11]. Although these percentages showcase comparability between the two contexts, the causes for these numbers vary. In terms of Canada, poverty is linked to factors such as access gaps and nutrition [12]. When it comes to Pakistan, it reflects systemic deficits in basic services that negatively impact the health system within the country, leading to higher disease and mortality rates.

Moreover, the Gini coefficient demonstrates an unequal distribution of income and consequently its effect on health. A study done in Canada showcases that inequalities rose during the 1980s–1990s but have since stabilized over ‘the past 25 years’ with most citizens being able to access core services, owing to higher national income and well-established public infrastructure [13]. However, low-income groups and indigenous populations remain disadvantaged. In Pakistan, the district data highlights rising income inequality as well as education gaps, particularly in rural areas [14]. This comparison underscores that reducing health inequities requires more than narrowing income distribution. In Pakistan, the challenge arises from structural inequalities in education and resources. While in Canada, inequalities occur due to wage stagnation and opportunity gaps. Addressing both contexts requires tailored, multisectoral policy responses that align with broader development and social protection goals.

### 3.3. Intersecting Gender, Indigenous, and Regional Health Inequities

Canada and Pakistan show significant gender inequities, yet the forms they take vary, influenced by the social and cultural dynamics of each country. In Canada, despite government frameworks and policies aimed at closing the gender gap and working toward Sustainable Development Goal 5 (promoting gender equality), gender inequities persist [15]. These are reported more specifically in women from low-income households and Indigenous groups. A report by the Organization for Economic Co-operation and Development (OECD) suggested that Indigenous women are over three times more likely to report spousal violence. In addition, Indigenous women face challenges in educational attainment, accessing healthcare, and experience a larger wage gap, with Indigenous women employed full-time earning 26 percent less than non-Indigenous men [16].

Due to long-standing socioeconomic disadvantages rooted in colonization, residential schools, forced relocation, and systemic discrimination, Indigenous groups in Canada—the First Nations, Inuit, and Métis—experience significantly poorer health outcomes than the general population. In 2015, 12% of non-Indigenous, non-racialized, non-newcomer children lived in poverty, compared to 47% of Status First Nations children [17]. These inequities are directly linked to reduced life expectancy. Inuit men live over 15 years less than the national average and have higher rates of chronic diseases such as diabetes, heart disease, and respiratory illness. Age-standardized diabetes prevalence is 17% among First Nations, 7% among Métis, and 5% among non-Indigenous people [18]. Mental health issues are also more prevalent, with higher rates of depression, substance use disorders, and suicide among Indigenous youth.

In contrast, in Pakistan, the poverty level intensifies the impact of gender norms, more directly limiting women’s ability to obtain healthcare. Gender inequities are deeply rooted in societal norms and cultural practices. Surveys and research have consistently highlighted that a majority of Pakistanis (approximately 63%) view women as being mistreated, facing discrimination within households and society at large, as well as experiencing high rates of gender-based violence and restricted autonomy. Moreover, studies indicate that approximately 70% of individuals living in poverty are women, making them the most vulnerable members of the community [19]. These entrenched gender disparities, combined with poverty and limited access to education and healthcare, exacerbate women’s social and economic marginalization, reinforcing a cycle of inequality that undermines overall health and development outcomes in Pakistan [20].

Regional health outcomes in Pakistan also vary due to inequities in infrastructure, economic development, and healthcare access. Urban centers like Islamabad, Karachi, and Lahore benefit from better facilities and skilled professionals, while rural areas, especially Balochistan, Khyber Pakhtunkhwa, and parts of Sindh, face poor infrastructure, underfunded systems, and workforce shortages. Conflict and political neglect further impair healthcare in regions like Khyber Pakhtunkhwa. The maternal mortality ratio is 154 per 100,000 nationally but rises to 298 in Balochistan and 224 in Sindh, showing pronounced gender-based inequities [21]. Rural women face barriers to maternity care from cultural norms, mobility restrictions, and a lack of trained staff. Urban slums in cities like Karachi and Lahore also face high disease risks such as cholera, dengue, and tuberculosis due to poor sanitation and limited access to clean water.

Together, these patterns of inequity in Canada and Pakistan highlight how gender, ethnicity, and geography intersect to shape access to healthcare and overall well-being. While Canada’s inequities are more strongly tied to systemic discrimination and the legacy of colonization affecting Indigenous groups, Pakistan’s challenges are rooted in poverty, patriarchal norms, and uneven regional development. Both contexts underscore the need for targeted, equity-focused health policies that address structural inequalities to promote inclusive and sustainable health outcomes.

### 3.4. Analytical and Contextual Analysis of the Current Policy Options

The framework of how the policies implemented in both Canada and address health inequities in access to healthcare has been analyzed based on the following: healthcare coverage and effectiveness in providing access to low-income populations, targeted policy implementation and success in addressing barriers; accessibility and financial constraints, and the role of social determinants such as living conditions, education, and employment on health outcomes.

In Pakistan, key policies addressing health inequities caused by poverty include strides toward universal health coverage. The Sehat Card, a government-funded health insurance program, aims to provide quality healthcare free of cost to the needy and has proven effective in offering access to care with good quality and affordability [22]. Efforts to expand Universal Health Coverage [UHC] include extending basic health units to rural, impoverished areas. Additionally, policies targeting education, housing, and nutrition have been supported by civil societies and NGOs like the Edhi Foundation, which offers women empowerment programs and microloans for small businesses [23].

In Canada, aside from the Medicare health insurance system that provides healthcare without upfront payment, other policies include community-level interventions like the Alberta health benefits program, which offers healthcare to low-income Albertans needing prescription medication or prenatal support [24]. Other programs cover dental, eye care, medication, and medical supplies, not included in Medicare. Canada also continues to address poverty-related health determinants through policies ensuring high literacy, safe housing, and clean food and water, supported by strict regulations.

The effects of national poverty reduction strategies on health outcomes are demonstrated through case studies from both countries. In Pakistan, the 2019 *Ehsaas Program* is a comprehensive multi-sectoral initiative that includes cash transfers via the Benazir Income Support Program, nutrition initiatives for women and children, and health insurance coverage. It also emphasizes social protection and human capital development. While it has improved access to healthcare and nutrition, it faces challenges with equitable distribution, rural outreach, political stability, beneficiary identification, awareness, and administrative efficiency. It reflects Pakistan’s recognition of the poverty-health link but depends on targeted, long-term execution for full impact.

In Canada, the *Canada Child Benefit* (*CCB*), implemented in 2016, significantly reduced child poverty from 15% in 2012 to 9% in 2017 [25]. The tax-free benefit for low- and middle-income families improved child health by reducing financial stress, improving nutrition, and increasing access to preventive care. However, health inequalities persist, especially for Indigenous populations, where chronic boil-water advisories, poor housing, and limited healthcare access affect communities like Attawapiskat.

Similarly, regional inequities exist in Pakistan. The *Lady Health Workers* (*LHW*) program in rural Sindh and Punjab provides vaccinations, maternal education, and basic preventive care but faces issues such as low pay, inconsistent training, and difficulty reaching remote areas.

These case studies show that while national policies are crucial, success depends on effective local implementation, continued support, and adaptation to the specific needs of the disadvantaged.

## 4. Discussion

### 4.1. Theoretical Frameworks for Tackling Multidimensional Conditions

This analysis uses the following frameworks to highlight health inequities in access to healthcare or health services as a multidimensional and multifaceted condition shaped by broader social determinants and to assess equity-focused policies.

The WHO Social Determinants of Health (SDH) Framework defines health as shaped by “the conditions in which people are born, grow, work, live, and age” [2]. Housing, education, and food security interact with poverty to worsen inequities, as seen in both Canada and Pakistan.

The Walt and Gilson Policy Triangle (1994) [26] examines actors, process, content, and context. Canada’s policies show welfare-state consistency, while Pakistan’s programs operate in a less stable, resource-limited setting. Medicare is state-led; the Sehat Card relies on NGOs and donors, reflecting differing governance.

HiAP stresses integrating health into all policy areas. Canada’s “Opportunity for All” links poverty reduction with health equity. Pakistan’s goals align with the Sustainable Development Goals (SDGs), aiming to improve health through housing, education, and employment policies.

Sen’s Capability Approach frames poverty as a deprivation of freedoms essential to health [27]. Health both enables and results from human capabilities. Canada’s universal care improves access; Pakistan’s Sehat Card aims similarly but faces barriers like inequality and weak infrastructure. This approach centers justice and freedom in understanding poverty-related health gaps.

### 4.2. Policy Impact Evaluation, Gaps, and Insights for Poverty Alleviation and Health Equity

Pakistan faces multiple gaps in the implementation of health and poverty-related policies to improve health and reduce the effects of poverty. For example, when it comes to the Sehat program, high political fragmentation and high levels of corruption along with the constant changing of governmental leaders, have caused a lack of ongoing implementation of the Sehat card policy throughout various provinces of the country. Further, a lack of awareness of the program is also a major cause of the initiative not being fully utilized. Moreover, even though the country is pushing policies to spread healthcare to rural sectors, these areas still remain experiencing high poverty rates and low access to healthcare, with even the healthcare workforce opting to work in urban areas compared to rural ones [28]. This inequity worsens the well-being of those who already struggle to pay for basic needs in rural areas. Moreover, there is a lack of vital policy implementation in the country in the first place. For example, there is a lack of policies to provide healthcare funding, which hampers the development of infrastructure and the provision of essential services. Additionally, there is a lack of focus on preventive health policies, overlooking the importance of addressing social determinants of health, such as education and sanitation, which are crucial for reducing health inequities. These factors in Pakistan’s policy implementation altogether increase health inequities caused by poverty. Although causality cannot be statistically established from secondary data, triangulation of policy and outcome trends provides plausible interpretive links between poverty reduction and improved equity.

In contrast, Canada faces comparatively fewer gaps. Some of which include the fact that while Medicaid does cover major healthcare expenses, it does not extend to certain primary care services [29] such as eye and dental care or medication, which can be especially problematic for impoverished individuals who may not be able to afford other insurance plans that can cover these expenses. Furthermore, there is a lack of consistency in implementing community and individual programs or policies. For instance, social housing programs to provide communities with safe housing at lower costs are known hubs for violence and lack of productivity, leading to even more poverty and creating a harmful environment for individuals’ health. Moreover, inequitable access to healthcare persists when it comes to people living in low-income areas of the country; individuals from these areas tend to have less access to screening and diagnosis services and experience higher mortality rates compared to those in more affluent regions.

### 4.3. Policy Recommendations

Building on this analysis and theoretical framework, the following recommendations are suggested.
For Pakistan

1. Enact a National Health Equity Act: Legislate universal healthcare as a constitutional right, prioritizing marginalized and rural populations to ensure accountability and reform continuity beyond political cycles.

2. Expand and universalize the Sehat Sahulat Program, scaling the program nationwide with simplified enrollment and expanded eligibility (e.g., informal workers, women-headed households, persons with disabilities). Transition to universal coverage for the bottom 60% by 2030.

3. Launch a Rural Health Strategy (2025–2035): Develop rural health centers and incentivize medical staff to serve remote areas via scholarships and bonuses. Additionally, deploy mobile units to isolated regions to increase coverage.

4. Integrate Preventive Health into Poverty Programs: Embed sanitation, nutrition, maternal-child health, and education into all poverty alleviation efforts to address root causes of health inequities.

5. Mandate Poverty and Health Impact Assessments (PHIA): Require all major policies to undergo PHIA to ensure they reduce, not worsen, health inequities—mainstreaming the HiAP approach.

6. Establish a Commission on Social Determinants of Health: Create a multisectoral body to align strategies across health, finance, education, sanitation, and housing.For Canada

1. Guarantee Universal Pharmacare and Dentalcare by 2030: Extend Medicare to cover prescription drugs, dental care, vision care, and mental health services to eliminate access barriers.

2. Improve Access in Underserved Areas: Boost funding for community health in low-income, Indigenous, and rural communities, ensuring primary and mental healthcare access.

3. Implement Indigenous Health Sovereignty: Secure long-term funding and co-management of services, restoring Indigenous health knowledge and governance.

4. Launch a Health Equity Monitoring System: Create a real-time, disaggregated data platform to inform equity-driven policies using Capability Approach metrics.

5. Reform Social Housing with Employment/Education Links: Revise housing policy to include job training and education support to reduce poverty-linked health risks.

6. Embed Health Equity in Economic Planning: Incorporate equity indicators into budgeting and infrastructure planning to align national growth with reduced inequities.

### 4.4. Limitations

This study relies primarily on secondary data and published, peer-reviewed literature; therefore, the findings are inherently constrained by potential publication bias, data quality variation, and limitations in the comparability of datasets across countries. To mitigate these issues, the authors applied a triangulation approach using multiple global and national data sources, including peer-reviewed studies and official databases to cross-verify evidence. In addition, all included studies underwent independent appraisal by the research team to ensure accuracy and consistency. While these measures enhanced the study’s validity and robustness, the authors acknowledge that future research incorporating primary data and longitudinal analysis would provide deeper empirical insight into evolving health inequities.

## 5. Conclusions

This comparative analysis reveals that inequities in access to healthcare remain a central determinant of health outcomes in both Pakistan and Canada, though they manifest differently. In Pakistan, inequities are worsened by political instability, limited funding, and weak rural healthcare infrastructure, despite programs like the Sehat Card and NGO efforts. In Canada, universal healthcare reduces financial barriers, yet low-income and Indigenous populations still lack access to dental care, prescription drugs, and essential services. The findings underscore that true health equity requires addressing poverty itself as a fundamental health intervention since socioeconomic deprivation underpins disparities in access and outcomes across both contexts. The comparative perspective of this study highlights that while Pakistan’s challenges stem from systemic underinvestment and service gaps, Canada’s inequities persist through structural exclusion within an otherwise well-resourced system. For Pakistan, this means investing in rural infrastructure, ensuring policy continuity, and expanding coverage. For Canada, this requires extending public coverage to dental, pharmaceutical, and mental health care while improving access for marginalized groups. Both must adopt equity-driven, evidence-based reforms that link poverty reduction directly to health outcomes and uphold health as a fundamental right. These findings inform future policy design by highlighting that reducing poverty is inseparable from advancing health equity. Across both contexts, inequities are not singular but intersecting and are shaped by overlapping dimensions of gender, geography, and income. Recognizing these intersections is essential for designing equity-driven health policies that address the compounded disadvantages experienced by women, rural populations, and Indigenous communities. Grounded in the WHO Social Determinants of Health framework and Sen’s Capability Approach, the findings demonstrate that inequities in healthcare access are deeply rooted in structural deprivations that limit individual freedoms and opportunities for well-being. Applying these frameworks highlights that poverty reduction and equitable policy design are mutually reinforcing processes that expand people’s capabilities to live healthy lives. Thus, advancing health equity requires addressing not only income disparities but also the broader social and political determinants that shape the conditions of health. Further research should integrate longitudinal and primary data to assess how post-2025 poverty reduction programs alter health access patterns.

## Data Availability

No new data were created or analyzed in this study. Data sharing is not applicable to this article.

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
