# Peer review of "A Comparative Policy Analysis of Health Inequities in Access to Healthcare Across Low- and High-Income Contexts: The Cases of Pakistan and Canada"

_ijerph, 2025, doi:10.3390/ijerph22111735_

Round 1
Reviewer 1 Report
Comments and Suggestions for Authors
Comments
Inequality of income and opportunity has been a problem with no end in sight for decades. Material and monetary poverty come in addition as contributing factor. Poor health is an inevitable and clear result of this socially ungrateful process.
Suggestions
The existing literature (methodology followed in the article) is always insufficient, although no topic is exhausted. Criticizing scientific articles is even more difficult, in their selection but mainly in discernment. Poverty Rate, Gini Coefficient, Health Expenditure, Life Expectancy, Infant Mortality, Maternal Mortality and Human Development Index, as treated in the document (Table 1), are not enough.
Comparing two extreme cases is like making a suggestion to political decision-makers, who don't read scientific papers (but should). This is in addition to the main aspect the authors avoid: the political and administrative organization, education and historical performance of Pakistan and Canada. Regardless of the recommendations, especially concerning to Pakistan, the measures proposed in the article will not be implemented in practice. The same applies to Canada, but at a much lower level. There is no optimal model anywhere.
The conclusions, beyond the methodology, are weak and of little substance. It would be better to first present a separate article for each country and then another with comparisons, based on the possible comparison of methodology and data analysis.
Comments on the Quality of English LanguageI am not from a country whose official language is English.
Reviewer 2 Report
Comments and Suggestions for Authors
Thank you for granting me the opportunity to review this article, which aims to analyze the income-related healthcare inequities in Pakistan and Canada. The authors present their arguments clearly, but the analysis lacks the rigor required to robustly derive the research conclusions. There are several issues that the authors should meticulously address:
- This study is based on a comparative analysis approach. However, the authors did not explicitly articulate the rationale for selecting Pakistan and Canada as analytical cases in the introduction section. Why are these two countries considered representative examples of low-income and high-income nations, respectively? Furthermore, in terms of health service systems, what specific aspects render these two countries comparable? The authors should provide clear justifications for these selections to strengthen the relevance and coherence of their comparative analysis.
- The reference list indicates that the majority of cited sources are policy documents rather than recent empirical research articles. Therefore, the authors are encouraged to systematically examine the disparities in access to health services between low-income and high-income populations. Drawing upon existing scholarly evidence, a critical analysis is warranted regarding the extent to which health policies have mitigated income-related health inequities. In addition, the study should place particular emphasis on exploring whether income-related health inequalities exhibit heterogeneous patterns across countries with differing income levels. Currently, the “3. Results" section primarily describes health policies in the two countries, but lacks a comprehensive discussion of recent empirical findings from the academic literature.
- Table 1 presents seven analytical indicators; however, the underlying purposes of these indicators are not elaborated. In addition, the differences between the two countries with respect to each indicator, as well as the associated health implications, are not examined. Although the study is framed as a comparative analysis, the discussion primarily consists of parallel descriptions of the two cases, without systematically comparing their similarities and differences within a coherent theoretical framework.
- The conclusion of this paper underscores that poverty reduction should be recognized as a form of health intervention, a perspective that has gained broad consensus in the academic community. The authors are encouraged to further refine the novel insights generated through comparative analysis to strengthen the scholarly contribution of this work.
- This paper seeks to investigate the relationship between poverty and inequality in access to health services. However, the Results and Discussion sections primarily present descriptive accounts of the historical and contemporary contexts of poverty, health services, and health policies in the two countries, without establishing a clear causal link between these factors based on empirical evidence. Consequently, the manuscript would benefit from additional literature review and analytical depth to adequately address its central research question.
In summary, although the topic of this manuscript is attractive, the paper requires a Major Revision to beef up its theoretical framing and explanation of results etc.
Reviewer 3 Report
Comments and Suggestions for Authors
***According to the abstract, this study employs a grounded approach (Grounded Theory?). Please justify the process of Grounded Theory that was adopted in this study.
***This study employed systematic thematic analysis and synthesis. Typically, the PRISMA method is used for systematic review and analysis of 37 publications. Please add the systematic review process to data collection and analysis. If this study did not employ the PRISMA method in meta-analysis, please add the study's limitations and recommendations for further studies.
***Please add selection criteria along with sampling techniques. There are two steps of sampling. The first step is to select Pakistan and Canada as the case study for comparison by purposive sampling. The second step is to select 37 papers for this comparison. Please add details.
***According to the methodology, please add the evaluators for paper selections. If only the authors were considered, it could be considered as biased without respondents in this study. Please add the study's limitations and recommendations for further studies that should be conducted for surveys.
***The health inequities in access should be compared in the same dimensions first.
According to the abstract's results, there is a comparison in different dimensions.
(In Pakistan, chronic underinvestment, rural-urban gaps, inadequate infrastructure, and political instability exacerbate inequities in access to healthcare. Limited coverage, ineffective administrative processes, and gaps in rural healthcare delivery impede growth despite encouraging programs like the Sehat Card and the Ehsaas Program. Conversely, Universal healthcare in Canada has lowered financial obstacles to access, but low-income and Indigenous communities are still impacted by service gaps, particularly in dental care, pharmacare, and mental health. Although child poverty rates have been significantly reduced by programs like the Canada Child Benefit, Indigenous children continue to endure disproportionate health risks. Findings underscore a need for equity-driven changes: Pakistan must expand rural health infrastructure and legislate health equity, while Canada should extend coverage to essential but excluded services. Both countries must prioritize poverty alleviation as a health intervention, integrating justice, sustainability, and accountability to advance global health equity).
It is noted that the comparison is in different ways and dimensions. Moreover, the poverty rate is different in different years: Pakistan in 2018 and Canada in 2022. Please ensure that it can be compared well.
***Lines 34-35: Health equity is the outcome of the sustainability approach. This, in Lines 34-35, should be: Both countries must prioritize poverty alleviation as a health intervention, integrating justice to advance global health equity for sustainability (social, economic, and environmental sustainability). If the authors would like to mention sustainability, please review the links between health equity and sustainability in three dimensions.
***Line 258 shows 4.3. Policy Recommendations: Sometimes, the experts' opinions based on interviews are recommended for further studies on respondent inclusion.
***The topic is about health inequities in access to healthcare: a comparative policy analysis. Please ensure the conclusion regarding the comparison in 'Inequities in Access to Healthcare' or 'Comparative Policy Analysis.'
It could be more comprehensive based on theories/variables about policies affecting health equity.
***Please add more scholarly papers (for a total of 30) in 2024-2025 based on Scopus, WOS, and Google Scholar from reputed journals.
Comments on the Quality of English LanguagePlease check grammar.
Round 2
Reviewer 1 Report
Comments and Suggestions for Authors
Comments
The literature review was reviewed and improved, becoming more robust when compared to the research on the indicators cited by the sources considered (first version of the article). The literature review was indeed important in improving the new document (authors' response 1). See the mention of the "WHO Social Determinants of Health framework" and the approach to Sen's thinking.
Also crucial was the inclusion in the article the indicators “gender inequities” and “regional inequities,” which did not exist in the initial version, as well as the broadening of the metrics of the indicators addressed in the original version (“income-based metrics”).
My criticism of the initial version has now been taken into account, because it was impossible to treat Pakistani reality as if it were Canadian reality at the same point of the article’s structure. It was essential to consider the separation of the two realities, which the authors did (response 2). Thus, the article is clearer to read and understand.
The results and conclusions, of course, had to be modified. They were also separated in their writing and structure within the article (Sections 3.2 and 3.3).
Comments on the Quality of English Language I am not a native speaker of an English-speaking country.Author Response
We sincerely thank the reviewer for their valuable feedback and constructive comments throughout the review process. Your insights and recommendations have been instrumental in helping us refine and strengthen the manuscript. We are very pleased that the revised version now meets the publication standards. The authors truly appreciate the time and effort invested in reviewing our work and look forward to seeing the paper published and contributing to the field.
Reviewer 2 Report
Comments and Suggestions for Authors
The authors have provided targeted modifications in response to the previously raised issues. Therefore, this paper now meets the publication standards.
Author Response
We sincerely thank the reviewer for their valuable feedback and constructive comments throughout the review process. Your insights and recommendations have been instrumental in helping us refine and strengthen the manuscript. We are very pleased that the revised version now meets the publication standards. The authors truly appreciate the time and effort invested in reviewing our work and look forward to seeing the paper published and contributing to the field.
Reviewer 3 Report
Comments and Suggestions for Authors
***Reviewer's comment 5: It is noted that the comparison is in different ways and dimensions. Moreover, the poverty rate is different in different years: Pakistan in 2018 and Canada in 2022. Please ensure that it can be compared well.
Authors' Response 5: We thank the reviewer for this valuable observation. In response, we have revised the manuscript to ensure that the comparison between Pakistan and Canada is made along consistent dimensions of health inequities, including financial barriers, rural–urban disparities, service coverage gaps, and vulnerable populations such as children and Indigenous communities. We have also clarified in the Results and Abstract sections that while the data points originate from slightly different years due to data availability (Pakistan 2018, Canada 2022), the comparison focuses on trends and relative disparities rather than absolute simultaneous values. This brings good evidence that this area of research requires further research that focuses on time series or chronological research to examine the topic in both countries over the years. We indicated this significant evidence in the revised manuscript, especially in the discussion and conclusion sections, to be considered. Although data on this topic over the year across countries is limited due to reporting cycles, the analysis aligns indicators to the most recent nationally reported estimates.
Comment to Authors' response 5: We have also clarified in the Results and Abstract sections that while the data points originate from slightly different years due to data availability (Pakistan 2018, Canada 2022), the comparison focuses on trends and relative disparities rather than absolute simultaneous values.
Please add a note that the comparison between Pakistan 2018 and Canada 2022 may not be valid due to different years and data availability (please do not add the word slightly different).
***Other comments are addressed and revised accordingly.
Author Response
We sincerely thank you for your valuable comments and suggestions.
Please find our responses (in red) to the comment raised by the reviewer.
Thanks for considering our paper, and we look forward to taking this revised and strengthened work forward.
Reviewer comment: Comment to Authors' response 5: We have also clarified in the Results and Abstract sections that while the data points originate from slightly different years due to data availability (Pakistan 2018, Canada 2022), the comparison focuses on trends and relative disparities rather than absolute simultaneous values.
Please add a note that the comparison between Pakistan 2018 and Canada 2022 may not be valid due to different years and data availability (please do not add the word slightly different).
Author's Response: We thank the reviewer for the additional clarification. We have now addressed this point in the revised manuscript by updating and aligning the data sources for both countries to ensure a more comparable time frame. Specifically, for Pakistan, the poverty data used in the study was collected between 2018 and 2019, as referenced in [11] Saddique, R., Zeng, W., Zhao, P., & Awan, A. (2023). Understanding multidimensional poverty in Pakistan: implications for regional and demographic-specific policies. Environmental Science and Pollution Research, 1–16. For Canada, the poverty estimate is now drawn from [10] Conde, A., Ferdinands, A. R., & Mayan, M. (2023). Working hard or hardly working? Who are Canada’s working poor? Journal of Poverty, 27(5), 351–373, which reports data from 2019. This update ensures a more valid comparison between the two countries.